# The Statistical Thermodynamics of Generative Diffusion Models: Phase Transitions, Symmetry Breaking, and Critical Instability

**DOI:** 10.3390/e27030291

**Published:** 2025-03-11

**Authors:** Luca Ambrogioni

**Affiliations:** Donders Institute for Brain, Cognition and Behaviour, Radboud University, 6525 XZ Nijmegen, The Netherlands; luca.ambrogioni@donders.ru.nl

**Keywords:** generative diffusion, statistical physics, phase transitions, spontaneous symmetry breaking

## Abstract

Generative diffusion models have achieved spectacular performance in many areas of machine learning and generative modeling. While the fundamental ideas behind these models come from non-equilibrium physics, variational inference, and stochastic calculus, in this paper we show that many aspects of these models can be understood using the tools of equilibrium statistical mechanics. Using this reformulation, we show that generative diffusion models undergo second-order phase transitions corresponding to symmetry breaking phenomena. We show that these phase transitions are always in a mean-field universality class, as they are the result of a self-consistency condition in the generative dynamics. We argue that the critical instability arising from these phase transitions lies at the heart of their generative capabilities, which are characterized by a set of mean-field critical exponents. Finally, we show that the dynamic equation of the generative process can be interpreted as a stochastic adiabatic transformation that minimizes the free energy while keeping the system in thermal equilibrium.

## 1. Introduction

Generative modeling is a subfield of machine learning concerned with the automatic generation of structured data such as images, videos, and written language [1]. Generative diffusion models [2], also known as score-based models, form a class of deep generative models that have demonstrated high performance in image [3,4], sound [5,6,7], and video generation [8,9]. Diffusion models were first introduced through analogy with non-equilibrium statistical physics. The fundamental idea is to formalize generation as the probabilistic inverse of a *forward stochastic process* that gradually turns the target distribution ϕ(y) into a simple base distribution such as Gaussian white noise [2,10]. Recently, several works have suggested that many of the dynamical properties of generative diffusion models can be understood using concepts such as spontaneous symmetry breaking [11,12,13] and phase transitions [13,14]. These theoretical and experimental results suggest a deep connection between generative diffusion and equilibrium phenomena.

In this paper, we outline a conceptual reformulation of generative diffusion models in the language of equilibrium statistical physics. We begin by defining a family of Boltzmann distributions over the noise-free states, which are interpreted as (unobservable) microstates during the diffusion process. In this picture, the Boltzmann weights are provided by the conditional distributions of the noiseless data given the noisy state. We obtain a self-consistent equation of state for the system which corresponds to the fixed-point equation of the generative dynamics. Moreover, we show that generative diffusion models can undergo second-order phase transitions of the mean-field type, corresponding to the generative spontaneous symmetry breaking phenomena first discussed in [11] and further studied in [13,14,15]. Finally, we show that this mean-field theory can be seen as the thermodynamic limit of a multi-site system of coupled replicas. Based on this result, we derive a variant of the generative diffusion equations as the Brownian dynamics of a ‘particle’ coupled to a large densely-connected system of replicated microstates, which offers a possible generalization of diffusion models beyond mean-field theory.

## 2. Contributions and Related Work

The main novel theoretical contributions of this paper are in its characterization of mean-field critical phase transitions in generative diffusion models and their extension beyond mean-field theory, which is provided in Section 9. While this paper contains novel results, its aim is also pedagogical, as we wish to provide a self-consistent introduction for physicists to the study of generative diffusion. As such, we report known formulas and results from the literature, including the analysis scheme used in [13,16] for the analysis of memorization phenomena and the equivalence results for modern Hopfield networks in [17]. While using the same random energy methods, the memorization analysis we provide in Section 11 is slightly different from the results in [13,16], as it does not use a signal-to-noise argument and is not restricted to ‘typical’ points. Several of these formulas can also be found in recent works on stochastic localization [18,19], which has been shown to offer an elegant generalization of generative diffusion processes [20,21,22]. In particular, the Boltzmann distributions in Equation (Equation 14) are equivalent to the tilted distributions in [18].

## 3. Preliminaries on Generative Diffusion Models

The goal of diffusion modeling is to sample from a potentially very complex target distribution ϕ(y), which we model as the initial boundary condition of a (forward) stochastic process that removes structure by injecting white noise. In order to simplify the derivations, we assume the forward process to be a mathematical Brownian motion. Other forward processes are more commonly used in the applied literature, such as the variance-preserving process (e.g., a non-stationary Ornstein–Uhlenbeck process) [4]. However, most of the qualitative thermodynamic properties are shared between these models. The mathematical Brownian motion is defined by the following Langevin equation:(1)$$x\left(t+dt\right)=x\left(t\right)+\;\sigma \;w\left(t\right)\sqrt{dt}$$
where dt is an infinitesimal increment, σ is the instantaneous standard deviation of the stochastic input, and w(t) is a standard Gaussian white noise process. Because this is a Brawnian motion, the probability of $x_{t}$ given $x_{0}$ at time *t* is provided by(2)$$p_{t}\left(x_{t}|x_{0}\right)=N\;\left(x_{t};x_{0},I\sigma^{2}t\right).$$
Using this formula, the marginal probabilities defined by Equation (Equation 1) (with ϕ(y) as the initial boundary condition) can be expressed analytically as follows:(3)$$p_{t}\left(x\right)=\frac{1}{\sqrt{2\pi t\sigma^{2}}}E_{y∼\phi }\;\left[e^{-\frac{{∥x-y∥}_{2}^{2}}{2t\sigma^{2}}}\right]$$
where the expectation is taken with respect to the target distribution ϕ(y). A generative model can then be obtained by “inverting” Equation (Equation 1). The inverse equation is(4)$$x\left(t-dt\right)=x\left(t\right)-\sigma^{2}\nabla \log p_{t}\left(x\right)+\;\sigma \;w\left(t\right)\sqrt{dt}\;,$$
which can be shown to provide the same marginal distributions in Equation (Equation 3) if the process is initialized with appropriately scaled white noise [23]. The function $\nabla \log p_{t}\left(x\right)$ is known in the literature as the score. If the score is available for all values of x and *t*, then we can sample from ϕ(y) by integrating Equation (Equation 4) using numerical methods.

### Training Diffusion Models as Denoising Autoencoders

While the score of the target distribution is generally not available analytically, a deep network can be trained to approximate it from a large set of training examples [10]. We refer to such a network as a vector valued function f(x(t),t). Deep networks are parameterized by a large number of weights and biases; however, as we are not interested in the details of the specific parameterization here, we report the functional loss:(5)$$L\left[f(\cdot ,\cdot )\right]=\int_{0}^{t_{\mathrm{end}}}E_{y,x(t)}\;\left[\frac{1}{2}{∥\frac{x(t)-y}{\sqrt{t\sigma^{2}}}-f\left(x\left(t\right),t\right)∥}_{2}^{2}\right]d\Psi \left(t\right)\;$$
where Ψ(t) is a cumulative distribution with support in $\left(0,t_{\mathrm{end}}\right)$ and x(t) is sampled conditional on y using the propagator of the forward Langevin equation. Note that $x(t)-y/\sqrt{t\sigma^{2}}$ is simply the total noise added up to time *t*, which implies that the network learns how to predict the noise corrupting the input data. The score function $\nabla \log p_{t}\left(x\right)$ can then be obtained from the optimized network $f^{*}=\mathrm{argmin}\;L\left[f\right]$ using the following formula [23]:(6)$$\nabla \log p_{t}\left(x\right)\approx -{\left(t\sigma^{2}\right)}^{-1/2}f^{*}\left(x\left(t\right),t\right).$$
In other words, the score is proportional to the optimal estimate of the noise given the noise-corrupted state. Therefore, after the network is trained to minimize Equation (Equation 5), synthetic samples can be generated by sampling $x(t_{\mathrm{end}})$ from the boundary noise, computing the score using Equation (Equation 6), and backwards-integrating Equation (Equation 4) using numerical methods. An example of this generative dynamics for a network trained on natural images is shown in Figure 1.

## 4. Preliminaries on the Curie–Weiss Model of Magnetism

In this section, we review the well known Curie–Weiss model of magnetism [24], which as we will see is deeply related to the statistical physics of generative diffusion. Consider a system of *N* coupled binary spins $s_{J}\in \left\{-1,1\right\}$ governed by the following Hamiltonian function:(7)$$H_{\mathrm{cw}}\left(\left\{s_{i}\right\}\right)=-J\sum_{(i,j)\in \Omega }s_{i}s_{j}-h\sum_{i}s_{i}$$
where Ω is a set of couplings, *J* is a coupling weight, and *h* is an external magnetic field that is static compared to the time scale of the fluctuations of the microstates $s_{i}$.

The thermodynamic behavior of this model can be studied by evaluating the partition function(8)$$Z\left(T,h\right)=\sum_{\left\{s_{i}\right\}}e^{-\frac{1}{T}H_{\mathrm{cw}}\left(\left\{s_{i}\right\}\right)},$$
where the sum is over all possible configurations of spins. Under most coupling structures, the sum in the partition function cannot be easily evaluated in close form. A tractable approximation can be obtained by rewriting the Hamiltonian as follows:(9)$$\begin{array}{cc} H_{\mathrm{cw}}\left(\left\{s_{i}\right\}\right) & =-\sum_{i}s_{i}\left(J\sum_{j}s_{j}+h\right)=-\sum_{i}s_{i}\left(Jdm+h\right) \end{array}$$
where *m* is the average magnetization(10)$$m=\frac{1}{d}\sum_{i=1}^{d}s_{i},$$
with *d* as the number of neighbors having a given spin. While this is a simple rewriting of the original energy function, we can use this form to obtain a tractable approximation by assuming that for a temperature *T*, *m* is equal to the thermal average of the spin: $m\left(T\right)={\langle m\rangle }_{T}$, which is assumed to be static when compared with the timescale of the microstates $s_{i}$. Using this assumption, we can compute the partition function of a spin by summing over its two possible states:(11)$$Z\left(T,h,m\right)=e^{-\frac{Jd}{T}\left(m+h\right)}+e^{\frac{Jd}{T}\left(m+h\right)}=2\cosh\;\left(\frac{Jd}{T}\left(m+h\right)\right).$$
To find the correct value of *m*, we need to solve a self-consistent mean-field equation:(12)$$m\left(T,h\right)=-T\frac{\partial }{\partial h}\log Z(T,h)=\tanh\;\left(\frac{Jd}{T}\left(m(T,h)+h\right)\right).$$
This equation has either one or two stable solutions depending on the values of *T* and *h*. For h=0, the system undergoes a critical phase transition at $T_{c}=Jd$, where a single stable solution m(T,h)=0 ‘splits’ into two stable solutions with either positive or negative magnetization. This results in so-called *spontaneous symmetry breaking*, as the observed statistics of the system spontaneously ‘pick’ one of the two options, resulting in an apparent violation of the original flip symmetry.

## 5. Diffusion Models as Systems in Equilibrium

We can now reformulate generative diffusion models in the language of statistical physics. The starting point for any model in statistical mechanics is the definition of the relevant microstates. In statistical physics, the microstate of a system is usually assumed to be an unobservable quantity. Given that we can observe a noise-corrupted data $x^{*}\left(t\right)$, the most obvious unobservable quantity of interest in a diffusion model is the noise-free initial state $x^{*}\left(0\right)=y^{*}$. The next step is to define a Hamiltonian function on the set of microstates. We can do this by considering the conditional probability of the data y given a noisy state $x_{t}$. This probability can be expressed using Bayes’ theorem:(13)$$p\left(y|x,t\right)=\frac{p_{t}\left(x_{t}|x_{0}\right)\phi \left(y\right)}{p_{t}\left(x\right)}=\frac{1}{p_{t}\left(x\right)\sqrt{2\pi t\sigma^{2}}}e^{-\frac{{∥x-y∥}_{2}^{2}}{2t\sigma^{2}}+\log\phi (y)}$$
where we have used Equations (Equation 2) and (Equation 3). We can now rewrite this probability distribution as(14)$$p\left(y|x,t\right)=\frac{1}{Z(x,t)}e^{-H(y;x,t)}\;,$$
which we can interpret as a Boltzmann distribution over the microstates y with the Hamiltonian(15)$$H\left(y;x,t\right)={\left(t\sigma^{2}\right)}^{-1}\left(\frac{1}{2}{∥y∥}_{2}^{2}-x\cdot y\right)-\log\phi (y)$$
and partition function(16)$$Z\left(x,t\right)=\int e^{-H(y;x,t)}dy.$$
The statistical properties of this ensemble determine the score function, which can be expressed as a Boltzmann average(17)$$\nabla \log p_{t}\left(x\right)=\beta \left(t\right)\left({\langle y\rangle }_{t,x}-x\right),$$
where $\beta \left(t\right)={\left(t\sigma^{2}\right)}^{-1}$ and(18)$${\langle y\rangle }_{t,x}=\int y\;\frac{e^{-H(y;x,t)}}{Z(x,t)}dy.$$
Intuitively, this equation tells us that the score vector directs the system towards the posterior average ${\langle y\rangle }_{t,x}$. As we shall see, studying the thermodynamics determined by these weights allows us to understand several important qualitative and quantitative features of the generative dynamics. For example, as we show in later sections, after a ‘condensation’ phase transition the score will only depend on a small number of data points, which can be detected by studying the concentration of the weights on a subexponential number of microstates. The thermodynamic system defined by Equation (Equation 14) does not have a true temperature parameter; however, the quantity $\sigma^{2}t$ plays a very similar role to the temperature in classical statistical mechanics. Moreover, in the Hamiltonian provided by Equation (Equation 15), the dynamic variable x is analogous to the external field term in magnetic systems, which can bias the distribution of microstates towards patterns ‘aligned’ in its direction. We can imagine x as being a “slower” thermodynamic variable that interacts (adiabatically) with the statistics of the microstates.

### 5.1. Example 1: Two Deltas

Most of the complexity of the generative dynamics comes from the target distribution ϕ(y). However, simple toy models can be used to draw general insights that often generalize to complex target distributions. A simple but informative example is provided by the following target:(19)$$\phi \left(y\right)=\frac{1}{2}\left(\delta (y+1)+\delta (y-1)\right)$$
where *y* is equal to either −1 or 1 with probability 1/2. Assuming the binary constraint, this results in the following diffusion Hamilton:(20)$$H\left(y;x,t\right)={\left(t\sigma^{2}\right)}^{-1}\left(\frac{1}{2}y^{2}-xy\right)-\log2$$
and the partition function reduces to a sum over the two states(21)$$Z\left(x,t\right)=\frac{1}{2}\left(e^{\beta (t)/2+\beta (t)}+e^{\beta (t)/2-\beta (t)}\right)=e^{\beta (t)/2}\cosh\left(\beta (t)x\right),$$
where $\beta \left(t\right)={\left(\sigma^{2}t\right)}^{-1}$. Apart from the $e^{\beta /2}/2$ factor, this expression is identical to the Curie–Weiss partition function in Equation (Equation 11), with the external field *h* replaced by the noisy state *x*. The magnetization can be obtained from the derivative of the free energy with respect to *x*:(22)m(x,t)=logZ(x,t)=tanh(β(t)x),
which is identical to the expression in the Curie–Weiss model, as the extra term does not depend on *x*. Using this result, we can now write the score function as(23)$$\nabla_{x}\log p_{t}\left(x\right)=\beta \left(t\right)\left(\tanh(\beta (t)x)-x\right).$$
The vector field determined by this function is visualized in Figure 2 together with its fixed points. It can be seen that an initial stable fixed point $x^{*}=0$ loses stability at $\beta (t_{c})=1$, where it bifurcates into two paths of fixed points. As we shall see in later sections, this bifurcation can be described as a spontaneous symmetry-breaking phase transition.

### 5.2. Example 2: Discrete Dataset

In real applications, generative diffusion models are trained on a large but finite dataset $D=\{y_{1},\cdots ,y_{N}\}$. Sampling from this dataset correspond to the target distribution(24)$$\phi \left(y\right)=\frac{1}{N}\sum_{j=1}^{N}\delta \left(y-y_{j}\right)\;.$$
If the data points are all normalized such that their norms are equal to one, this results in the partition function(25)$$Z\left(x,t,N\right)=\frac{e^{-\beta (t)/2}}{N}\sum_{j=1}^{N}e^{\beta \left(t\right)\;x\cdot y_{j}}.$$
This partition function plays a central role in the random-energy analysis of the model, which can be used to study the finite sample thermodynamics.

### 5.3. Example 3: Hyper-Spherical Manifold

Because datasets are always finite, in practice every trained generative diffusion model corresponds to the discrete model outlined in the previous subsection. However, fitting the dataset exactly leads to a model that can only reproduce the memorized training data. Instead, the hope is that the trained network will generalize and interpolate the samples, thereby approximately recovering the true distribution of the sampled data. Very often, this distribution will span a lower-dimensional manifold embedded in the ambient space.

A simple toy model of data defined in a manifold is the hyper-spherical model introduced in [11]:(26)$$Z\left(x,t,d\right)=\frac{e^{-\beta (t)/2}}{V(d-1)}\int_{S(d-1)}e^{-\beta (t)x\cdot y}dy$$
where S(d−1) denotes a d−1-dimensional hypersphere centered at zero with volume V(d−1). The “two delta” model is a special case of this model for an ambient dimension equal to one. As shown in subsequent sections, this data distribution is very tractable in the infinite dimensional (i.e., thermodynamic) limit, as it converges to a distribution of normalized Gaussian variables, which removes the quadratic terms in the Hamiltonian.

### 5.4. Example 4: Diffused Ising Model

While most of the formulas presented in this manuscript have a very close analogy with formulas in statistical physics, there are some subtle interpretative differences that could create confusion in the reader. To clarify these issues, we discuss the *diffused Ising model*, which can provide a bridge between the two views. Consider a diffusion model with a target distribution supported on *d*-dimensional vectors y with entries in the set {−1,1}. The log-probability of the target distribution is defined by the following formula:(27)$$\log\phi (y)=\frac{1}{2T}\sum_{j\neq k}y_{j}y_{k}W_{jk}+c$$
where W is a symmetric coupling matrix, *T* is a temperature parameter, and *c* is a constant. Up to constants, this is of course the log-probability of an Ising model without the external field term. From Equation (Equation 15), up to constant terms, we obtain the following Hamiltonian for the diffusion model:(28)$$H\left(y;x,t,T\right)=-\beta \left(t\right)x\cdot y-\frac{1}{2T}\sum_{j\neq k}y_{j}y_{k}W_{jk}$$
which is almost identical to the Hamiltonian of an Ising model coupled to a location-dependent external field x. Nevertheless, the quantity $\beta \left(t\right)={\left(t\sigma^{2}\right)}^{-1}$, which we loosely interpreted as “inverse temperature”, does not divide the coupling part of the Hamiltonian, which results in radically different behavior. In fact, $t\sigma^{2}$ only modulates the susceptibility to the field term; therefore it does not radically alter the phase of the model, which depends on the Ising temperature parameter *T*. Instead, the interesting phase transition behavior of diffusion models is a consequence of the self-consistency relation in Equation (Equation 32), which characterizes the branching of the fixed points of the generative stochastic dynamics. From the point of view of statistical physics, Equation (Equation 32) can be seen as the result of a mean-field approximation in which the average magnetization is coupled to the external field. However, it is important to keep in mind that in a diffusion model this mean-field approach does not represent the coupling between individual sites, which Equation (Equation 28) shows to instead be statistically coupled by the interaction terms in the Hamiltonian. Instead, it can be seen as an idealized mean-field interaction between infinitely many copies of the whole system. In general, the value of *T* changes the properties of the diffusion model as the system transitions from its low-temperature phase to its high-temperature phase. The dependency of the diffusion dynamics on this transition have been studied in [12].

## 6. Free Energy, Magnetization, and Order Parameters

Using our interpretation of $\beta \left(t\right)={\left(t\sigma^{2}\right)}^{-1}$ as the inverse temperature parameter, we can define the Helmholtz free energy as follows:(29)$$F\left(x,t\right)=-\beta^{-1}\left(t\right)\log Z(x,t).$$
The expected value of the pattern y given x can then be expressed the gradient of the free energy with respect to x:(30)$${\langle y\rangle }_{t,x}=-\nabla F\left(x,t\right).$$
This formula suggests an analogy between diffusion models and magnetic systems in statistical physics. The noisy state x can be interpreted as an external magnetic field, which induces the state of “magnetization” ${\langle y\rangle }_{t,x}$. In this analogy, a diffusion model is magnetized when its distribution is biased towards a subset of the possible microstates.

In physics, the ‘external field’ variable x is usually assumed to be controlled by the experimenter. On the other hand, in generative diffusion models x is a dynamic variable that, under the reversed dynamics, is itself attracted towards ${\langle y\rangle }_{t,x}$ by the drift term:(31)$$\nabla \log p_{t}\left(x\right)=\beta \left(t\right)\left({\langle y\rangle }_{t,x}-x\right).$$
In other words, if we ignore the effect of the dispersion term, the state of the system is driven towards self-consistent points where x is equal to ${\langle y\rangle }_{t,x}$. Therefore, it is interesting to study the self-consistency equation(32)m(h,t)=−∇F(m(h,t)+h,t),
which defines the self-consistent solutions m(t) where the state is identical to the expected value. In this equation, we introduce a perturbation term h which allows us to study how systems react to perturbations. For h=0, the equation can be equivalently re-expressed as the fixed-point equation of the reversed drift:(33)$$\nabla \log p_{t}\left(m\left(0,t\right)\right)=0.$$
For t→∞, this equation admits the single “trivial” solution $m={\langle y\rangle }_{0}$, where ${\langle \cdot \rangle }_{0}$ denotes the expectation with respect to the target distribution ϕ(y). In analogy with magnetic systems, we can interpret m(h,t) as an order parameter and this equation as a thermodynamic equation of state. This analogy suggests that m(h,t) can be interpreted as a ‘spontaneous magnetization’ of the system. From this point of view, we can conceptualize the generative process as a form of self-consistent spontaneous symmetry breaking in which the system aligns with one of the many possible target points. In the following sections, we formalize this insight by characterizing the critical behavior of this system.

Readers familiar with statistical physics will recognize that Equation (Equation 32) is formally identical to the self-consistency conditions used in the mean-field approximation, where the external field term in one location is assumed to be determined by the magnetization of all other locations. However, it is important to note that in the case of a diffusion model this self-consistent coupling is not approximate, as it is a natural consequence of the dynamics. Nevertheless, the formal analogy implies that the thermodynamics of generative diffusion models are formally identical to the thermodynamics of mean-field models.

### The Susceptibility Matrix

In the physics of magnetic systems, the magnetic susceptibility matrix determines the extent to which different magnetization components are sensitive to the components of the external magnetic field. Similarly, in diffusion models we can define a susceptibility matrix(34)$$\chi_{ij}\left(x,t\right)=\frac{\partial m_{i}}{\partial h_{j}}{|}_{h=0},$$
which tells us how sensitive the expected value is to changes in the noisy state x(t). The susceptibility matrix is helpful in interpreting the dynamics of the generative denoising process, as it provides information on how random fluctuations in each component of the state x(t) are propagated to the other components. For example, in the context of image generation, a random fluctuation of “green” at the bottom of an image can propagate to the rest, originating the image of a forest.

The susceptibility matrix can be re-expressed in terms of the connected correlation matrix (i.e., the covariance matrix) of the microstates under the Boltzmann distribution(35)$$\chi_{ij}\left(x,t\right)=-\beta \left(t\right)\left({\langle yy^{T}\rangle }_{x}-{\langle y\rangle }_{t,x}{\langle y^{T}\rangle }_{x}\right)=-\beta \left(t\right)C_{ij}\left(x,t\right).$$

We can now express the Jacobi matrix of the score function as follows:(36)$$J_{ij}\left(x\right)=\beta \left(t\right)\left(\chi_{ij}\left(x,t\right)-\delta_{j,k}\right)=-\beta \left(t\right)\delta_{j,k}-\beta {\left(t\right)}^{2}C_{ij}\left(x,t\right).$$

## 7. Phase Transitions and Symmetry Breaking

Spontaneous symmetry breaking occurs when the trivial solution for the order parameter branches into multiple solutions at some critical time $t_{c}$. This corresponds to the onset of multimodality in the regularized free energy $\tilde{F}\left(x,t\right)$, as visualized in Figure 3 for a “four deltas” model. In thermodynamic systems, this symmetry breaking corresponds to a (second-order) phase transition, which in this case can be detected by the divergence of several state variables around the critical point. The presence of one (or multiple) phase transitions in diffusion models depends on the target distribution ϕ(y). The simplest example is provided by the “two deltas” model, which corresponds to the generative diffusion process visualized in Figure 2b. The critical point can be obtained by studying the following self-consistency equation:(37)$$m=\tanh\left(\frac{m}{t\sigma^{2}}\right).$$
The solutions of this equation are shown in Figure 2a together with the gradient of the regularized free energy, where we can see the branching of the solutions and the singular behavior around a critical point. Equation (Equation 37) is identical to the mean-field self-consistency equation of an Ising model (i.e., a Curie–Weiss model), from which we can deduce that the critical scaling of this simple generative diffusion models shares its universality class. For example, by Taylor expansion of Equation (Equation 37) around $\sigma^{2}t_{c}=1$, we can see that(38)$$m∼{\left(-\tau \right)}^{1/2}$$
with $\tau =\sigma^{2}\left(t-t_{c}\right)$, which is valid for *t* smaller than $t_{c}$.

### Generation and Critical Instability

As shown in [11], spontaneous symmetry breaking phenomena play a central role in the generative dynamics of diffusion models. Consider the simple “two deltas” model. For $t>>t_{c}$, the dynamics undergo mean-reversion towards a unique fixed point m(0,t). Around $t=t_{c}$, the order parameter splits into thee “branches”: an unstable one corresponding to the mean of the target distribution, and two stable ones corresponding to the two target points. Importantly, the susceptibility defined in Equation (Equation 34) diverges at the critical point, implying that the system becomes extremely reactive to fluctuations in the noise. This instability is determined by the critical exponents δ and γ, and is defined by the relations(39)$$h_{j}\propto m_{j}^{\delta_{j}}\;$$
and(40)$$\chi_{ij}\propto {\left(-\tau \right)}^{\gamma_{ij}}.$$
Note that in the general case the critical exponents can be different for different coordinates and matrix entries; these divergences give rise to something we refer to as *critical generative instability*. We conjecture that the diversity of the generated samples crucially depends on the proper sampling of this critical region.

## 8. Generation as an Adiabatic Free Energy Descent Process

Thus far, we have characterized the thermodynamic state of diffusion model at time *t* by its Boltzmann distribution. The dynamics of the system can now be recovered as a form of (stochastic) free energy minimization:(41)$$x\left(t-dt\right)=x\left(t\right)-\beta \left(t\right)\nabla \tilde{F}\left(x,t\right)dt+\sigma w\left(t\right)\sqrt{dt}$$
where $\tilde{F}$ is the free energy plus a free potential term(42)$$\tilde{F}\left(x,t\right)=F\left(x,t\right)+V\left(x\right),$$
where $V\left(x\right)=\frac{1}{2}{∥x∥}_{2}^{2}$. This can be seen as a form of adiabatic approximation in which we obtain the dynamics of the ‘slow’ variable x by assuming that the system is maintained in thermal equilibrium along the diffusion trajectory. Symmetry breaking can now be detected as a change of shape in the regularized free energy, which transitions from a convex shape with a single global minimum to a more complex shape, potentially with several metastable points (see Figure 3). The reformulation of the dynamics in term of the gradient of the free energy allows us to interpret generative diffusion models as a kind of energy-based machine learning models [25], as discussed in [17,26]. The main difference is that the (free) energy is not learned directly but is instead implicit in the learned score function. This suggests a potential connection with the free energy principle in theoretical neuroscience, which is used to characterize the stochastic dynamics of biological neural systems [27].

## 9. Beyond Mean-Field Theory: A Multi-Site ‘Generative Bath’ Model

The results in the previous sections suggest that generative diffusion models can be seen as a mean-field limit of a model with replicated microstates on *K* ‘sites’ coupled through long-range interactions. We denote the microstate in the *j*-th site as $y^{j}$. Consider the following multi-site Hamiltonian:(43)$$H_{K}\left[y^{1:K};h\right]=-\beta \left(\frac{1}{2K}\sum_{j=1,k=1}^{K}y^{j}\cdot y^{k}-\sum_{j}^{K}{∥y^{j}∥}_{2}^{2}-\sum_{j}^{K}y^{j}\cdot h\right)-\sum_{j}\log\phi \left(y^{j}\right).$$

In the model, different replications of the microstates $y_{j}$ (i.e., the noise-free data) exert mutual attractive couplings. Generation can be seen as a spontaneous symmetry-breaking event that the system undergoes when the temperature (i.e., the time) decreases, since at low temperatures all of the microstates in all sites will align on the same pattern, resulting in a coherent observable average(44)$$\bar{y}=\frac{1}{K}\sum_{\mu }^{K}y^{\mu }.$$
As we will show, in the thermodynamic limit (K→0) the model converges to the self-consistent mean-field model discussed in the previous sections; this allows us to conceptualize the self-consistency condition implicit in the fixed-point equation of generative diffusion model as the result of an ideal multi-site coupling. This conceptualization opens the door for possible non-mean-field generalizations of generative diffusion characterized by short-range interactions or disordered generalizations with random interactions. However, it is not clear whether these extensions will have practical value.

### 9.1. Connection Between the Multi-Site Model and the Fixed-Point Structure of Diffusion Models

We now show that in the thermodynamic limit K→∞, the statistical behavior of the system described by the multi-site Hamiltonian in Equation (Equation 43) gives rise to the mean-field symmetry breaking that characterizes the dynamics of conventional generative diffusion models. Consider the partition function(45)$$Z_{K}\left(\beta \right)=\int \prod_{\mu =1}^{K}\phi \left(y^{\mu }\right)dy^{\mu }e^{\beta \left(\frac{1}{2K}\sum_{j,k}^{K}y^{j}\cdot y^{k}-\sum_{j}^{K}{∥y^{j}∥}_{2}^{2}\right)}.$$
The terms including the multi-site couplings can be expressed in terms of the magnetization:(46)$$\sum_{j,k}^{K}y^{j}\cdot y^{k}=N^{2}{∥m∥}_{2}^{2}$$
with(47)$$m=\frac{1}{K}\sum_{j=1}^{K}y^{j}.$$
Using this expression, we can rewrite the partition function as(48)$$Z_{K}\left(\beta \right)=\int dm\prod_{\mu =1}^{K}\phi \left(y^{\mu }\right)dy^{\mu }e^{\beta \left(\frac{1}{2}K{∥m∥}_{2}^{2}-\sum_{j}^{K}{∥y^{j}∥}_{2}^{2}\right)}\delta \;\left(Km-\sum_{j=1}y^{j}\right).$$
We can now use the Fourier representation of the delta function$$\delta \left(x\right)=\int \frac{d\omega }{{\left(2\pi \right)}^{d/2}}e^{i\omega \cdot x}\;,$$
which decouples the integrals with respect of each site, leading to the expression(49)$$Z_{K}\left(\beta \right)=\int \frac{d\omega }{{\left(2\pi \right)}^{d/2}}dm\;e^{K\left(\beta /2{∥m∥}_{2}^{2}+i\omega \cdot m+\log\psi (\omega ,\beta )\right)},$$
where the integrals over each site provide us with the function(50)$$\psi \left(\omega ,\beta \right)=\int e^{-\beta {∥y∥}_{2}^{2}-i\omega \cdot y}\phi \left(y\right)dy.$$
We can now evaluate the remaining integrals for K→∞ using the saddle point method, which results in the two conditions:(51)βm=−iω
and(52)$$im=-\nabla_{\omega }\log\psi \left(\omega ,\beta \right).$$
By combining these two equation, we obtain the self-consistency equation(53)$$m=\beta^{-1}\nabla_{m}\log\int e^{-\beta \left({∥y∥}_{2}^{2}-y\cdot m\right)}\;\phi \left(y\right)dy=-\nabla_{m}F\left(m,\beta \right),$$
which is identical to the fixed-point equation in Equation (Equation 32).

### 9.2. Brownian Dynamics in a ‘Generative Bath’

The Hamiltonian defined in Equation (Equation 44) specifies an equilibrium system that, in the thermodynamic limit, shares the same self-consistent criticality of the fixed-point equation of generative diffusion models. In this section, we derive from first principles a generative stochastic dynamics similar to the generative equation in Equation (Equation 4). The idea is to consider a ‘Brownian particle’ x(t) coupled to the multi-site system of microstates. We define the random force as(54)$$F\left(x,t\right)=\frac{1}{H}\sum_{\mu }^{H}\left(y^{\mu }-x\right),$$
where *H* is the number of sites coupled to x. In contrast to the distribution in Equation (Equation 14), we assume that x does not exert any effect on the equilibrium systems itself, which instead undergoes symmetry-breaking events due to its own internal coupling between sites. In other word, in this formulation the state x(t) is now passively controlled by the statistical fluctuations in the equilibrium system. If we assume that the force in Equation (Equation 54) is applied at each infinitesimal time interval and that its time scale is much faster that the motion of x(t), then the Brownian dynamics follow the (reversed) Langevin equation(55)$$x\left(t-dt\right)-x\left(t\right)=-\left({\langle \bar{y}\rangle }_{t,x\to 0}-x\left(t\right)\right)\frac{dt}{t}+\frac{1}{\sqrt{H}}B\left(x,t\right)w\left(t\right)\sqrt{\frac{dt}{t}},$$
where $B\left(x,t\right)=C^{1/2}\left(x\right)$ is a matrix square root of the pure state covariance matrix:(56)$$C\left(x\right)=\left({\langle \bar{y}{\bar{y}}^{T}\rangle }_{t,x\to 0}-{\langle \bar{y}\rangle }_{t,x\to 0}{\langle {\bar{y}}^{T}\rangle }_{t,x\to 0}\right)\;.$$
The 1/t scaling in the differential is introduced in order to have the reversed diffusion ends at the finite time t=0, which is equivalent to a logarithmic change of coordinate in the time variable. The Boltzmann expectation is taken with respect to the multi-site ensemble provided in Equation (Equation 44) in the limit of a vanishing external field aligned to x. This is done in order to isolate the appropriate ‘pure state’ from the Boltzmann average, since after a spontaneous symmetry breaking only one branch of the distribution should affect the particle. In fact, after a symmetry-breaking phase transition, the Boltzmann distribution splits into two or more modes corresponding to the possible states with broken symmetry (see [28]).

### 9.3. The Two Delta Model Revisited

In the “two deltas” model, up to constants, the Hamiltonian of the “generative bath” is just(57)$$H_{K}\left[y^{1:K};h\right]=-\beta \left(\frac{w}{2}\sum_{j,k}^{K}y^{\mu }y^{\nu }\right),$$
with the restriction that $y^{\mu }\in \left\{\pm 1\right\}$. This is simply the Hamiltonian of a fully connected Ising model with uniform coupling weights. Therefore, in the thermodynamic limit K→∞, the model reduces to the mean-field Curie–Weiss model that we have already discussed. In this case, the pure-state magnetization consists of the stable solutions $m_{t}$ of the self-consistency equation $m=\tanh\left(m+h\right)/\left(t\sigma^{2}\right)$, which is identically equal to zero for $t\sigma^{2}>1$ and has two branches that cannot be expressed in closed-form within the low-temperature regime. The instantaneous variance of the Brownian generative dynamics is provided by $T\partial m_{t}/\partial h$, which is equal to $1/(1-{\left(t\sigma^{2}\right)}^{-1})$ in the high-temperature phase and $T\left(1-m_{t}^{2}\right)/m_{t}^{2}$ in the low-temperature phase. Note that the variance diverges at $t\sigma^{2}=1$ due to the critical phase transition, and that it vanishes for t→0 as the system fully aligns in one of its two possible pure states.

## 10. Associative Memory and Hopfield Networks

We now move back to the standard mean-field formulation of generative diffusion and discuss its connection with associative memory networks. Associative memory networks are energy-based learning systems that can store patterns (i.e., memories) as metastable states of a parameterized energy function [29,30,31]. There is a substantial body of literature on the thermodynamic properties of associative memory networks [32,33,34]. The original associative memory networks, also known as *Hopfield networks*, are defined by the energy function $E\left(x\right)=\frac{1}{2}x^{T}Wx$ under the constraints of binary entries for the state vector. In a Hopfield network, a finite number of training patterns $y_{j}$ are encoded into a weight matrix $W=\sum_{j}y_{j}y_{j}^{T}$, which usually provides the correct minima when the number of patterns is on the order of the dimensionality. Associative memory networks can reach much higher capacity by using the exponential energy function [31,35,36]. For example, [37] introduce the use of the following function:(58)$$E\left(x\right)=-\beta^{-1}\log\left(\sum_{j}e^{\beta x\cdot y_{j}}\right)+\frac{1}{2}{∥x∥}_{2}^{2},$$
which can be proven to provide exponential scaling of the capacity and is related to the transformer architectures used in large language models [37]. By inspecting Equation (Equation 58), we can see that this energy function is equivalent to the regularized Helmholtz free energy of a diffusion model trained on a mixture of delta distributions [17]:(59)$$\phi \left(y\right)=\sum_{j}\delta \left(y-y_{j}\right)$$
which provides a free energy with the same fixed-point structure of Equation (Equation 58) at the zero temperature limit. Note that while the dynamics of a diffusion model do not necessarily act as an optimizer in the general case, the free energy is exactly optimized when ϕ(y) is a sum of delta functions, making the dynamics of the model exactly equivalent to the optimization of Equation (Equation 58) for β→∞. Given this connection, most of the results presented in this paper can be restated for associative memory networks. However, generative diffusion models are more general, as they can target arbitrary mixtures of continuous and singular distributions. As we show in the next section, the modern Hopfield Hamiltonian plays a crucial role in studying finite sample effects such as data memorization [16].

## 11. The Random Energy Thermodynamics of Diffusion Models on Sampled Datasets

As seen in the previous sections, we can define a Hamiltonian function using the target density ϕ(y), and consequently define an equilibrium thermodynamic system that captures the statistical properties of the corresponding generative diffusion dynamics. From the point of view of machine learning, this corresponds to a denoising network perfectly trained on an infinitely large dataset. Consequently, this analysis misses some important properties that arise when models are trained on finite datasets sampled independently from ϕ(y). In particular, the exact model cannot capture the phenomenon of memorization (overfitting), which refers to a model failing to generalize beyond its training set.

We can analyze this regime using the statistical physics of disordered systems, where the quenched partition function depends on *N* randomly sampled training points:(60)$$Z_{N}\left(x,\beta \left(t\right)\right)=\frac{1}{N}\sum_{j=1}^{N}e^{-\beta \left(t\right)\left(\frac{1}{2}{∥y_{j}∥}_{2}^{2}+x\cdot y_{j}\right)},$$
where $y_{j}∼\phi \left(y\right)$. Here, the idea is that we can understand the finite sample properties of diffusion models by studying how the partition function and other thermodynamic quantities fluctuate as a function of the random sampling. This is analogous to the physics of glasses, where the thermodynamic properties depend randomly on the (disordered) structure of a piece of material. By defining $E_{j}\left(x\right)=\frac{1}{2}{∥y_{j}∥}_{2}^{2}-x\cdot y_{j}$, we can rewrite this quenched partition function as a random energy model (REM):(61)$$Z_{N}\left(x,\beta \left(t\right)\right)=\frac{1}{N}\sum_{j=1}^{N}e^{-\beta \left(t\right)E_{j}\left(x\right)}$$
where the distribution of the energy levels $E_{j}\left(x\right)$ depends on the “external field” x. The thermodynamics of this model are well known for simple distributions over the energy, and can be studied in more complex cases using the replica method [28].

### Memorization as ‘Condensation’

The random-energy analysis of a diffusion model is very useful for studying the memorization phenomenon, where the diffusion trajectories collapse on the individual sampled datapoints instead of spreading to the underlying target distribution. In machine learning jargon, this phenomenon is known as overfitting. It is clear that a model perfectly trained on a finite dataset (i.e., a model that perfectly reproduces the empirical score) fully memorizes the data without any generalization for t→0. However, these over-fitted models can still exhibit generalization for finite values of *t*, as the noise level can be too high to distinguish the individual training points. It was recently discovered that in the thermodynamic limit, generalization and memorization are demarcated by a disordered symmetry-breaking [13,16]. This is a so-called condensation phenomenon, where the probability measures the transition from being spread out over an exponential number of configurations to being concentrated on a random small (sub-exponential) set. During condensation, the score is determined by a relatively small number of non-vanishing Boltzmann weights, which eventually direct the dynamics towards one of the training points.

Using our theoretical framework, we can find this ‘critical time’ $t_{\mathrm{cond}}$ by studying the random partition function provided in Equation (Equation 61) in the thermodynamic limit. This can be done by evaluating the ‘participation ration’(62)$$Y_{N}\left(x,t\right)=\frac{Z_{N}\left(x,2\beta \left(t\right)\right)}{{\left(Z_{N}\left(x,\beta \left(t\right)\right)\right)}^{2}},$$
where $1/Y_{N}\left(x,t\right)$ provides a rough count of the number of configurations with non-vanishing probability. In the REM theory, the participation ration can be used to detect a condensation phase transition, as in this case its value becomes identically equal to zero for $\beta >\beta_{c}$ in a non-analytic manner.

Consider the simple case where we sample $N=2^{M}$ data points from a distribution ϕ(y) which is uniform over a *d*-dimensional hypersphere with radius $r=\nu \sqrt{M/2d}$, where ν is a parameter that regulates the standard deviation of the data. In this case, up to an irrelevant constant shift, the random energy becomes(63)$$E_{j}\left(x\right)=-x\cdot y_{j}.$$
For high values of *d*, the distribution of $x\cdot y_{j}$ is approximately a centered Gaussian with variance ${∥x∥}_{2}^{2}\nu^{2}M/2$; as ϕ(y) is spherically symmetric, it does not depend on the direction of x. Therefore, we can approximately rewrite the expression as a standard REM model(64)$$Z_{N}\left(x,\beta \left(t\right)\right)=\frac{1}{2^{M}}\sum_{j=1}^{2^{M}}e^{-\tilde{\beta }{\tilde{E}}_{j}},$$
where ${\tilde{E}}_{j}∼N\left(0,M/2\right)$ and $\tilde{\beta }=\nu \beta \left(t\right){∥x∥}_{2}$. Note that this formulation is not exact at the thermodynamic limit, as it ignores the residual effect of non-Gaussianity in the distribution. A more rigorous analysis can be found in [16] in the context of modern Hopfield networks.

For the standard REM model, we know that the critical condensation temperature is at $\beta_{c}=2\sqrt{\log2}$ (see [28]), which leads to the critical time(65)$$t_{\mathrm{cond}}\left(x\right)=\frac{\nu {∥x∥}_{2}}{2\sigma^{2}\sqrt{\log2}}.$$
This formula was first derived in [16] for associative memory models. It can be shown that, at the limit M→∞, the expected participation ratio is(66)$$E\;\left[Y(x,t)\right]=\left\{\begin{array}{cc} 0, & \mathrm{if}\;\beta \left(t\right)\leq \beta_{c}\\ 1-\frac{\beta_{c}}{\beta (t)}, & \mathrm{if}\;\beta \left(t\right)>\beta_{c} \end{array}\right\}\;.$$
This allows us to estimate the number of data points with non-negligible weight in the score as n(x)≈1/Y(x,t). Note that $t_{c}$ does not predict the collapse on an individual data point, as that would require Y(x,t)≈1, which happens at an earlier time; however, it identifies a critical transition in the score function ∇logp(x) that demarcates the beginning of a memorization phase. In this phase, the system undergoes further symmetry-breaking phase transitions which break the symmetry between the data points with non-vanishing weights. The critical behavior of these transitions is analogous to the “two deltas” model.

## 12. Experimental Evidence of Phase Transitions in Trained Diffusion Models

The presence of one or more phase transitions in generative diffusion models is hard to prove theoretically for complex data distributions; however, symmetry breaking can be inferred experimentally from trained networks. For example, [11] showed that the generative performance of models trained on images stays largely invariant when the reversed dynamics are initialized before $t_{\mathrm{end}}$ up to a critical point, to the extent that the system is initialized using a Gaussian with a properly chosen mean vector and covariance matrix. This is consistent with our theoretical analysis, as the marginal distributions have a single global mode prior to the first phase transition and are approximated well by normal distributions. The effect of this form of symmetry breaking was further studied both experimentally and theoretically in [14], where it was found that generative diffusion models trained on natural images undergo a series of phase transitions corresponding to hierarchical class separations. This leads to low-level visual features emerging at earlier times of the diffusion process, while higher level semantic features emerge later. These results where further studied in [13], where the authors provided a series of analytic formulas for the critical times and verified the prediction experimentally on several image datasets. In particular, it was found that the timing of symmetry breaking corresponding to splits in different classes (e.g., images of dogs and cats) can be predicted based on the eigenvectors of the covariance matrix of the data. Together, these results strongly suggest that generative diffusion models undergo symmetry-breaking phase transitions under most realistic data distributions.

## 13. Conclusions

In this paper, we have presented a formulation of generative diffusion models in terms of equilibrium statistical mechanics. This allows us to study the critical behavior of these generative models during second-order phase transitions as well as the disordered thermodynamics of models trained on finite datasets. Our analysis establishes a deep connection between generative modeling and statistical physics, which in the future may allow physicists to study these machine learning models using the tools of computational and theoretical physics.

## Figures and Tables

**Figure 1 entropy-27-00291-f001:**
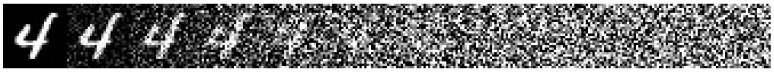
Generative process for a digit taken from the MNIST dataset.

**Figure 2 entropy-27-00291-f002:**
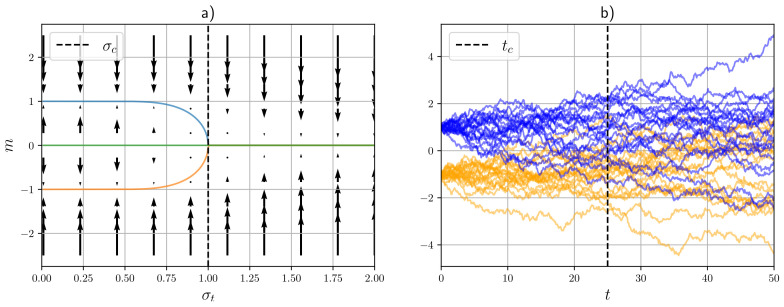
Visualization of a phase transition in a simple diffusion model (two deltas). (**a**) Order parameter paths and (regularized) free energy gradients; the dashed line denotes the critical value of $\sigma_{t}=\sqrt{t}\sigma_{0}$. (**b**) Forward process; the dashed line denotes the critical time. The color denotes the starting location of each particle.

**Figure 3 entropy-27-00291-f003:**
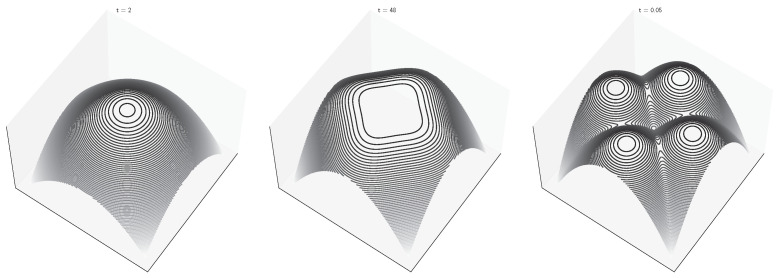
Negative free energy of a “four delta” 2D diffusion model for different values of the time variable. The target points are at (1,0), (0,1), (−1,0) and (0,−1).

## Data Availability

The original contributions presented in this study are included in the article. Further inquiries can be directed to the corresponding author.

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
