# Peer review of "The Statistical Thermodynamics of Generative Diffusion Models: Phase Transitions, Symmetry Breaking, and Critical Instability"

_entropy, 2025, doi:10.3390/e27030291_

Round 1
Reviewer 1 Report
Comments and Suggestions for Authors
I am not a specialist on machine learning (and generative modeling), so it is difficult to gauge the originality of this proposal. I enjoyed the connections with equlibrium statistical mechanics, although the text is somewhat rough, with a number os small mistakes. Even the example of the "two deltas" could have been explaned much better (also, I belive thare are some misprints). It would have been helpful to work out the explicit connections with the "Curie-Weiss model". Section 7, beyond the mean-field methods, is too sketchy and quite hard to understand.
Comments on the Quality of English LanguageExcept for some misprints, and minor mistakes (Ornstein, for example), the language seems allright.
Author Response
I wish to thank the reviewer for the insightful review. I agree that the original text had several rough and unclear parts, which I clarified and expanded in the revision. Here is a list of changes in the revised text:
- I made the text more understandable by adding a preliminary section on the Curie-Weiss model (Section 3) and by extending the treatment of the main ideas in Section 4 and the Two-deltas example in Section 4.1.
- I greatly expanded the section concerning generalization of diffusion beyond mean-field theory (Section 8). Now it includes a full derivation of the connection with the fixed-point equation of standard generative diffusion, which is achieved in the thermodynamic limit.
- I corrected several small errors and misprints throughout the text.
Reviewer 2 Report
Comments and Suggestions for Authors
The paper presents a link between generative diffusion models and equilibrium statistical mechanics, highlighting the role of phase transitions, symmetry breaking and criticality in generative diffusion models. The paper is well written, relevant for the intended audience and journal, and it contains interesting parallels between generative models and statistical mechanics. I recommend this paper for publication in Entropy, only after a few very minor corrections.
Typos and minor comments:
- P2 line 37: fist discussed
- P2 line 57: initial initial
- P2 line 61: do you mean Ornstein-Uhlenbeck process?
- P2 line 76: using a from a
- p7 line 213 would read better if change to: ... determines how sensitive the different magnetization components are to the ....
- Figure 2 is first referred to in the text at line 233, if I'm not mistaken. This is after figure 3 is referred to in the text. Either make a reference to Fig 2 close to where it is placed or move the figure closer to where it is first referenced.
- At times it is stated that t is like a temperature, other times it is \sigma t and at other times \sigma^2 t is identified as temperature. This gets a bit confusing, it is better to use one or the other and to be clear whether this is temperature or INVERSE temperature. For instance in line 126 the identification \beta = \sigma^2 t is made, while in line 161 and line 178 \beta is identified with (t \sigma^2)^{-1}. Which one is it? Is t temperature or inverse temperature? Please go through the text and provide a consistent definition of this, including in wording, by explicitly stating whether you are talking about temperature or inverse temperature.
- Also the critical point is not consistently used throughout the text. In the caption of Fig 2, it is stated as \sigma_t = \sqrt{t} \sigma_0, while in line 240 t_c = 1 and in line 339 the critical point is at \sigma^2 t = 1. Please make this internally consistent throughout the text
Author Response
I wish to thank you for the positive review and for the feedback on all the minor errors. I connected the errors in the revision.
Reviewer 3 Report
Comments and Suggestions for Authors
Please find our comments in the enclosed pdf file.

Author Response
I wish to thank the reviewer for the insightful review. I expanded several sections to make the paper more understandable for the non-expert. In particular, I included a preliminary section on the Curie-Weiss model (Sec.3) and expanded the presentation in Sec.4, including a more extensive treatment of the two-deltas model. To clarify what the novel contributions are, I expanded section 1. In summary, the main novel results are in the characterization of mean-field phase transitions in terms of the Curie-Weiss model and the generalization of diffusion models beyond the mean-field regime using the multi-site model. While the latter construction is entirely novel, the former is a re-formulation of the results in 11 and 13. Nevertheless, these previous works do not offer an explicit mean-field characterization of the phase transitions and do not cover important topics such as the critical exponents and the susceptibility matrix.
Round 2
Reviewer 1 Report
Comments and Suggestions for Authors
The additions on the Curie-Weiss model are still somewhat confusing.
Reviewer 3 Report
Comments and Suggestions for Authors
We believe the author addressed most of our concerns.